# Highly selective urea electrooxidation coupled with efficient hydrogen evolution

Guangming Zhan [1,2], Lufa Hu[1,2], Hao Li [1] ✉, Jie Dai [1], Long Zhao [1], Qian Zheng[1], Xingyue Zou[1], Yanbiao Shi [1], Jiaxian Wang[1], Wei Hou[1], Yancai Yao[1] ✉ & Lizhi Zhang [1] ✉

Electrochemical urea oxidation offers a sustainable avenue for $H_2$ production and wastewater denitrification within the water-energy nexus; however, its wide application is limited by detrimental cyanate or nitrite production instead of innocuous $N_2$. Herein we demonstrate that atomically isolated asymmetric Ni−O−Ti sites on Ti foam anode achieve a $N_2$ selectivity of 99%, surpassing the connected symmetric Ni−O−Ni counterparts in documented Ni-based electrocatalysts with $N_2$ selectivity below 55%, and also deliver a $H_2$ evolution rate of 22.0 mL h$^{-1}$ when coupled to a Pt counter cathode under 213 mA cm$^{-2}$ at 1.40 $V_{RHE}$. These asymmetric sites, featuring oxygenophilic Ti adjacent to Ni, favor interaction with the carbonyl over amino groups in urea, thus preventing premature resonant C═N bond breakage before intramolecular N−N coupling towards $N_2$ evolution. A prototype device powered by a commercial Si photovoltaic cell is further developed for solar-powered on-site urine processing and decentralized $H_2$ production.

Urea is a milestone molecule in modern chemistry because of its substantial contribution to agricultural, industrial, and medical prosperity and is also a crucial N-containing biomolecule generated through mammalian protein catabolism[1,2]. However, the overuse of urea in agriculture, in conjunction with its runoff and overwhelming load of urea exceeding 58 megatons annually in municipal treatment facilities (Supplementary Notes 1), poses a latent threat to both ecosystem stability and human health[3]. Uncontrolled discharge of urea leads to eutrophication, intensifying the blooms of algae and other aquatic plants. The hydrolysis of urea yields ammonia, which, if released into the atmosphere, accelerates acid rain occurrence[4]. Generally, urea is treated through adsorption, biodegradation, reverse osmosis, and chemical oxidation methods, which have high operational complexity and consume large amounts of energy[5]. Notably, urea stands out as a decent hydrogen carrier ($H_2$) with a gravimetric hydrogen content of 6.71 wt.%, fulfilling the present target as a reliable alternative to fossil fuels[6]. In the pursuit of transforming waste into valuable resources and achieving carbon neutrality, it is highly important to develop new strategies for urea-containing wastewater treatment and simultaneous $H_2$ production[7,8].

The electrochemical urea oxidation reaction (UOR) to innocuous $N_2$ offers a sustainable solution for urea-containing wastewater denitrification, taking advantage of its facile working conditions and high processing capacity[9,10]. More importantly, compared with the oxygen evolution reaction (OER), the UOR offers a thermodynamically favorable pathway for supplying electrons for the coupled cathodic hydrogen evolution reaction (HER) (Fig. 1a)[11]. The development of UOR has been restricted by its dependence on expensive rare metals such as ruthenium, platinum, or iridium[12]. Recently, several earth-abundant and cost-effective transition metals, particularly nickel (Ni) and its oxides (NiO, NiOOH) with urease-like catalytic centers, have exhibited attractive UOR performances[13–16]. However, these documented Ni-based electrocatalysts frequently convert urea to deleterious cyanate (NCO$^-$) and nitrite/nitrate (NO$_x^-$) species with $N_2$ selectivities below 55%[17,18], which might lower the stability of the integrated electrochemical system by poisoning the UOR anode and increase the environmental risk[19]. Structurally, urea contains two electron-donating amino (NH$_2$) groups and an electron-withdrawing carbonyl (C=O) group that collectively engage in resonance to yield two C═N bonds with ~30% double bond character[20,21]. For the prevalent Ni-based

[1]School of Environmental Science and Engineering, Shanghai Jiao Tong University, Shanghai, P. R. China. [2]These authors contributed equally: Guangming Zhan, Lufa Hu. ✉e-mail: hao_li@sjtu.edu.cn; yyancai@sjtu.edu.cn; zhanglizhi@sjtu.edu.cn

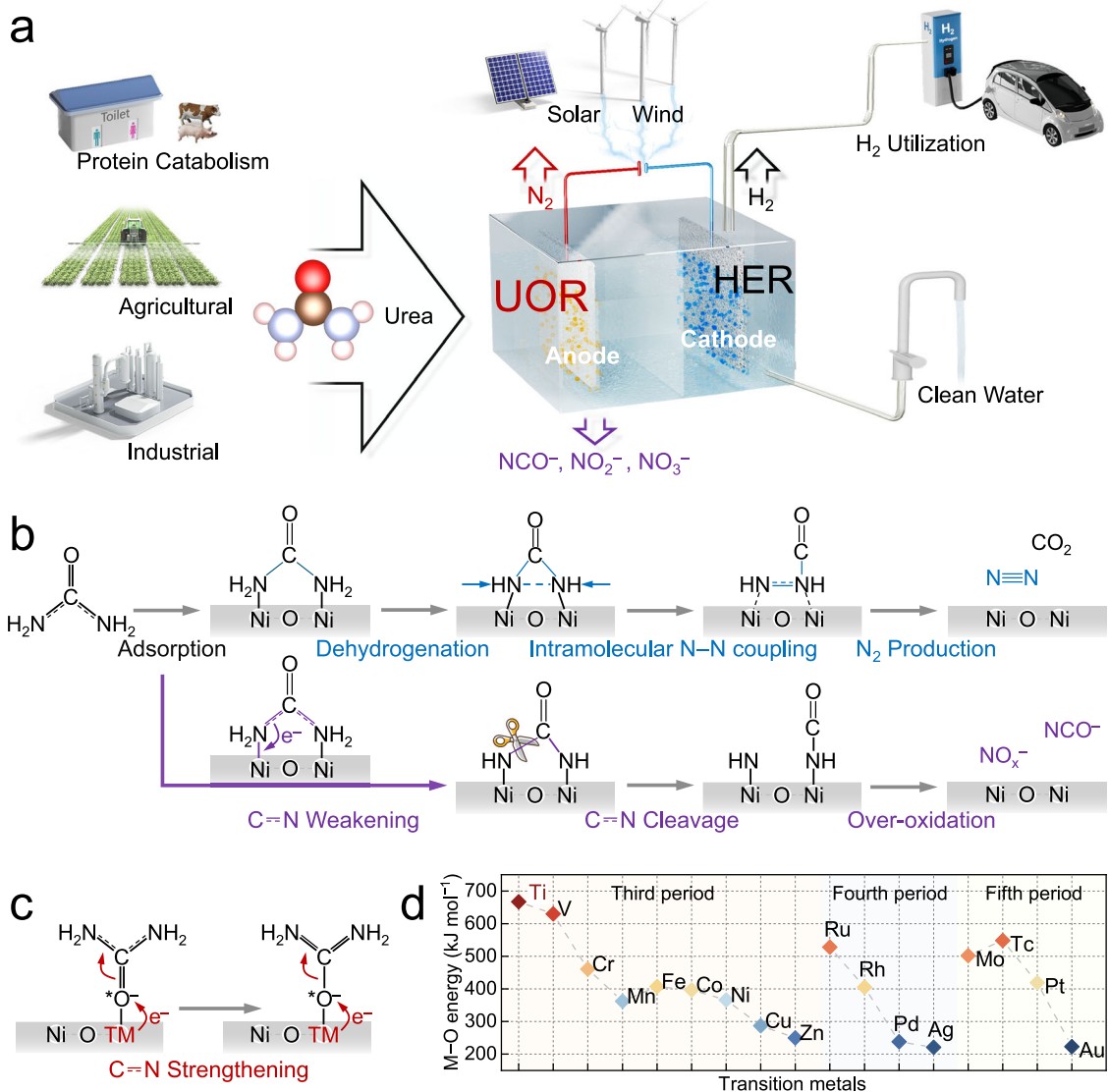

**Fig. 1 | Schematic illustration of selective UOR mechanisms and motivations.**
**a** Scenario of the UOR towards clean water and energy production. **b** The UOR pathways for $N_2/CO_2$ or $NO_x^-/NCO^-$ production on connected symmetric Ni–O–Ni sites. **c** Illustration of strengthening the resonant C═N bond of urea by donating electrons to the terminal O moiety of the C = O group. **d** The bond formation energies of O with Ti, Ni, and other common transition metals at 298.15 K, defined as the standard enthalpy change for the formation of a diatomic species between O and the transition metal.

catalysts under oxidative conditions, their connected symmetric Ni–O–Ni sites bind to urea in a symmetric bidentate-binuclear configuration via electron-donating $NH_2$ ends[22,23], blocking the resonance between the $NH_2$ and C = O groups and thereby weakening the C═N bond, as evidenced by its transition from a quasi double bond to a single bond[20]. Theoretically, $N_2$ formation proceeds through the pivotal intramolecular N–N bond coupling reaction during stepwise dehydrogenation of urea[24]. However, C═N weakening usually leads to its cleavage into cyanate ($NCO^-$) and $^*NH_x$ (where x = 1, 2), followed by their subsequent overoxidation to produce secondary $NO_x^-$ (Fig. 1b)[18]. Therefore, the cleavage of the C═N bond in urea should be avoided for the selective conversion of urea to $N_2$ during the electrochemical UOR, which might be achieved by donating electrons to the terminal O moiety of the C = O group to increase the electron accumulation on the resonant C═N bond (Fig. 1c).

The C═N bond electron density of urea can be increased by pairing the catalytic Ni center with an auxiliary metal site that exhibits pronounced oxygenophilic affinity, which would facilitate electron donation to the terminal O moiety of the C = O group[25–27]. A careful

assessment of the periodic table underscores the prominence of titanium (Ti) as an optimal choice owing to its bond formation energy of O (667 kJ mol$^{-1}$), which is significantly greater than that of Ni (366 kJ mol$^{-1}$), and other common transition metals (Fig. 1d)[28,29]. In this study, we endeavored to engineer atomically isolated Ni–O–Ti sites in an asymmetric configuration for electrochemical UOR into $N_2$ by affixing single Ni atoms onto a commercially available Ti foam to preclude the formation of connected symmetric Ni–O–Ni sites and allow adjacent basal Ti atoms for strong C = O→Ti interactions[26]. Theoretical calculations in combination with in situ Raman spectroscopy, Fourier transform infrared spectroscopy, and on-line mass spectrometry were employed to reveal the formation of asymmetric Ni–O–Ti sites, electronically strengthened C═N bonds in urea and kinetically facilitated intramolecular N–N bond coupling, as well as their contributions to the $N_2$ selectivity of the UOR. Furthermore, we conceptualized and engineered a prototype UOR–HER device by coupling a Ti foam monolithic electrode, which features atomically isolated asymmetric Ni–O–Ti sites, to a Pt cathode for efficient cathodic $H_2$ evolution. Powered by a commercial Si photovoltaic cell, this device

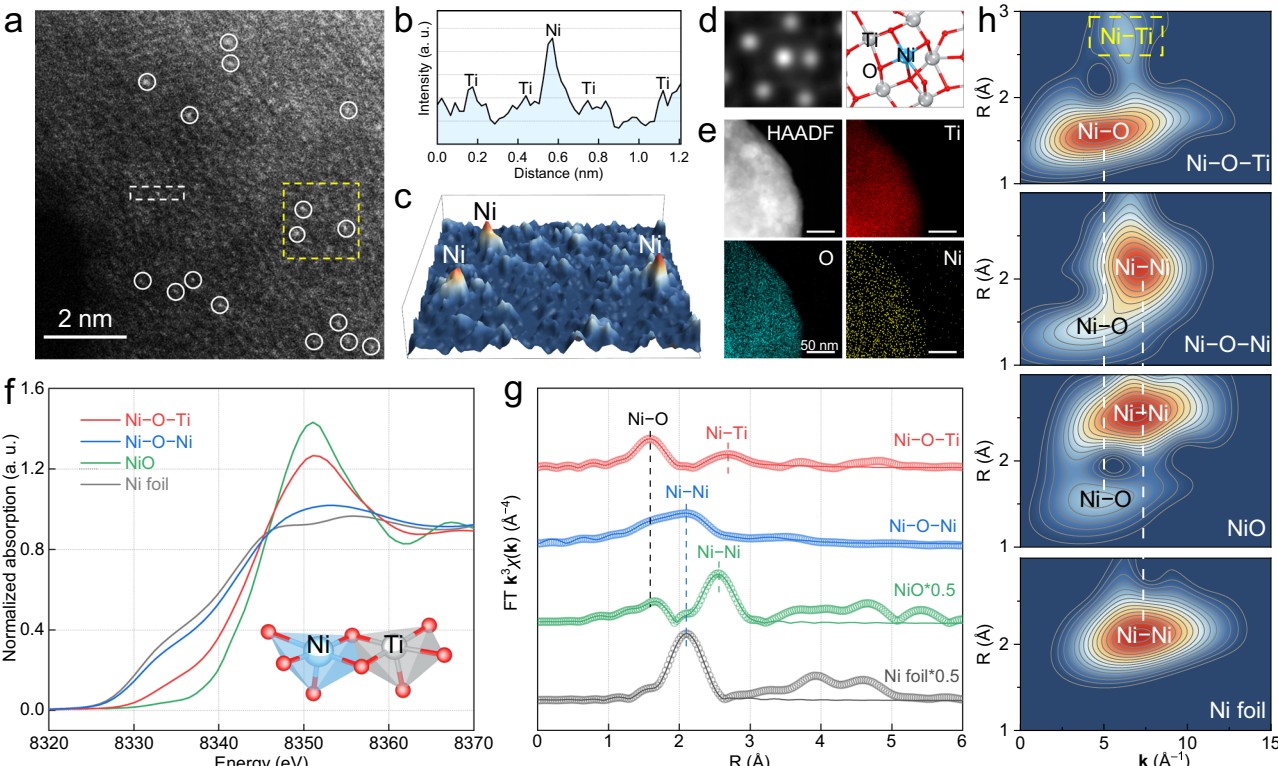

**Fig. 2 | Characterization of the atomically isolated asymmetric Ni–O–Ti sites.** **a** Atomic-resolution HAADF-STEM image of asymmetric Ni–O–Ti sites with white circles marking Ni atoms. **b** Corresponding line intensity profile taken along the white dashed rectangle, and (**c**) 3D surface intensity profile in the yellow dashed square box. **d** 2D black and white STEM simulation images on the basis of the Z contrasts of atomic columns and corresponding structure diagram of asymmetric Ni–O–Ti sites. **e** STEM elemental mapping. **f** Ni K-edge X-ray absorption near-edge structure spectra of the atomically isolated asymmetric Ni–O–Ti sites and connected symmetric Ni–O–Ni sites on Ti foam. **g** FT-EXAFS spectra and (**h**) wavelet transformed (WT) $k^3$-weighted $\chi(k)$-function of Ni K-edge of the asymmetric Ni–O–Ti sites and symmetric Ni–O–Ni sites. Ni foil and NiO were used as the standard samples.

can harness solar energy for on-site urine processing and decentralized fuel production.

## Results

### Materials characterization

The atomically isolated asymmetric Ni–O–Ti sites were constructed by anchoring single Ni atoms onto a commercial monolithic Ti foam. The key to the synthesis of asymmetric Ni–O–Ti sites is the in situ generation of coordinated unsaturated Ti with dangling bonds on the amorphous $TiO_x$ surface layer of the Ti foam under a reductive 5.0% $H_2$/Ar atmosphere (Supplementary Figs. 1, 2)[30]. These unsaturated Ti atoms could provide reliable immobilization sites for single Ni atoms via strong metal-support interactions. Aberration-corrected high-angle annular dark-field scanning transmission electron microscopy (HAADF-STEM) revealed that the Ni atoms were atomically dispersed on the surface of the $TiO_x$ (Fig. 2a), which was highlighted by the corresponding line and 3D surface intensity profile aligned with the STEM simulation (Fig. 2b–d). Complementary STEM-mapping and X-ray diffraction (XRD) revealed the uniform dispersion of Ni atoms without discernible Ni clusters or nanoparticles (Fig. 2e and Supplementary Fig. 3). Inductively coupled plasma–mass spectrometry (ICP-MS) analysis revealed a Ni loading of ~0.29 wt.% (Supplementary Table 1). For comparison, the connected symmetric Ni–O–Ni sites on the Ti foam anode were synthesized by increasing the Ni loading to 1.08 wt.% (Supplementary Fig. 4).

We then employed X-ray absorption near-edge structure (XANES) and extended X-ray absorption fine structure spectroscopy (EXAFS) to decipher the chemical state and coordination environment of atomically isolated asymmetric Ni–O–Ti sites. The absorption edge energy of Ni in the asymmetric Ni–O–Ti sites was between those of standard Ni

foil and NiO, suggesting the positively charged nature of the Ni atoms (Fig. 2f). In contrast, the absorption edge energy of connected symmetric Ni–O–Ni sites on Ti foam was similar to that on Ni foil, indicating of a more metallic state and consistent with the X-ray photoelectron spectroscopy (XPS) results (Supplementary Fig. 5). The Fourier-transformed EXAFS spectra of the asymmetric Ni–O–Ti sites displayed a dominant peak at 1.59 Å and a minor peak at 2.69 Å (Fig. 2g), which were attributed to Ni–O coordination in the first shell and Ni–Ti coordination in the second shell, respectively. The **k**-space of the minor peak located at 6.1 Å$^{-1}$ differed from the 7.0 Å$^{-1}$ **k**-space of the Ni–Ni coordination found in Ni foil, NiO, Ni(OH)$_2$, or symmetric Ni–O–Ni sites according to the wavelet-transform EXAFS (Fig. 2h and Supplementary Fig. 6). Notably, the peak for metallic Ni–Ni scattering at 2.09 Å in the asymmetric Ni–O–Ti sites was absent, highlighting the atomically dispersed nature of the Ni atoms. Quantitative structural analysis indicated that each Ni atom in the asymmetric Ni–O–Ti sites was coordinated with approximately five adjacent Ti atoms in the second shell, which were interconnected by five O atoms (Supplementary Figs. 7, 8 and Table 2). As a comparison, the Ni atoms within the Ni–O–Ni sites on the Ti foam and conventional Ni-based materials were exclusively coordinated with Ni to form connected symmetric Ni–O–Ni sites.

### Electrochemical performances

According to linear sweep voltammetry (LSV), the atomically isolated asymmetric Ni–O–Ti sites exhibited low potentials of 1.30 V and 1.33 V vs. reversible hydrogen electrode ($V_{RHE}$) to achieve laboratory- and industrial-level current densities of 10 mA cm$^{-2}$ and 100 mA cm$^{-2}$, respectively, while the connected symmetric Ni–O–Ni counterparts required slightly higher potentials (Fig. 3a and Supplementary

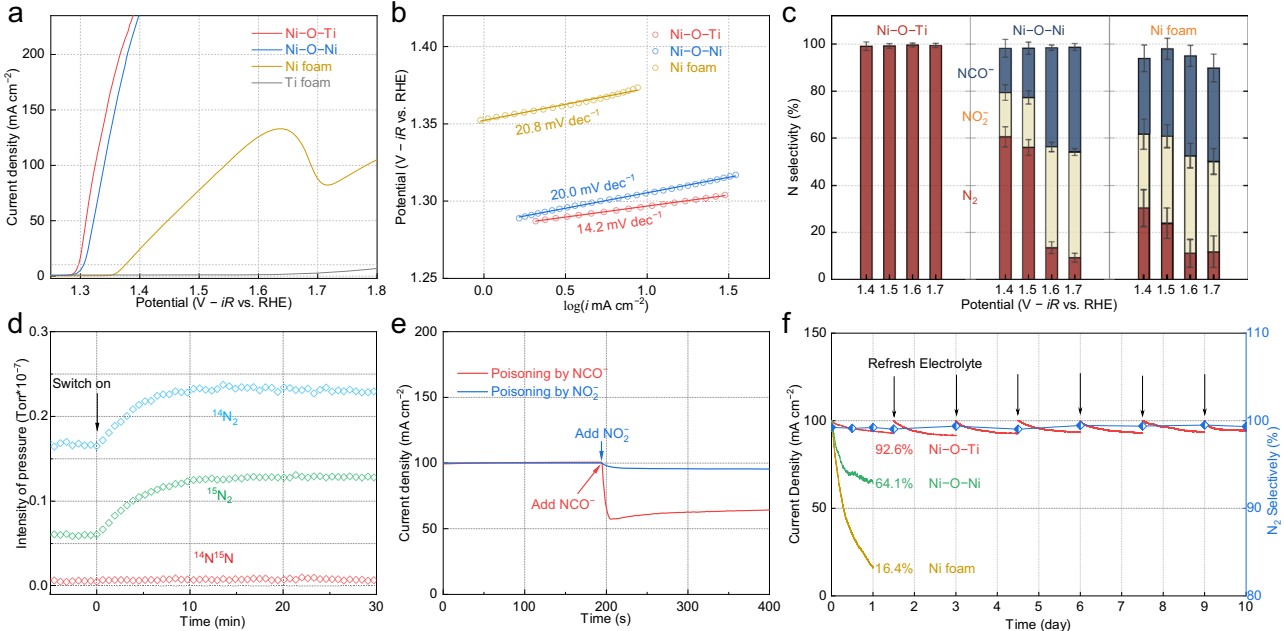

**Fig. 3 | Electrochemical performances of the atomically isolated asymmetric Ni–O–Ti sites. a** LSV curves with 85% *iR* correction and (**b**) Tafel plots of the atomically isolated asymmetric Ni–O–Ti sites and connected symmetric Ni–O–Ni sites on Ti foam in 1.0 M KOH containing 0.33 M urea with a stirring rate of 500 rpm and 5 mV s⁻¹. Freestanding Ti foam and Ni foam were used for comparison. The solution resistance is $2.1 \pm 0.1\,\Omega$. **c** The UOR selectivity of asymmetric Ni–O–Ti sites, symmetric Ni–O–Ni sites, and Ni foam as a function of the applied potential. Error bars represent the standard deviations derived from three distinct samples. **d** On-line MS for asymmetric Ni–O–Ti sites in 1.0 M KOH with 0.1 M CO($^{14}$NH$_2$)$_2$ and 0.1 M CO($^{15}$NH$_2$)$_2$. **e** The current density changes of asymmetric Ni–O–Ti sites after the introduction of 0.1 M NCO⁻ and 0.1 M NO$_2^-$ in 1.0 M KOH containing 0.33 M urea electrolyte. **f** Chronoamperometry measurements with the initial current density of 100 mA cm⁻² in 1.0 M KOH containing 1.0 M urea, and the solution resistance is $2.0 \pm 0.1\,\Omega$.

Figs. 9–12). For comparison, freestanding Ti foam was ineffective at catalyzing the UOR, while the Ni foam showed poor UOR activity, demanding considerably higher potentials of 1.38 and 1.54 $V_{RHE}$ to reach equivalent current densities of 10 mA cm⁻² and 100 mA cm⁻², respectively. Notably, the Ni–O–Ti sites had a substantially smaller Tafel slope (14.2 mV dec⁻¹) than did the Ni–O–Ni sites (20.0 mV dec⁻¹) and the Ni foam (20.8 mV dec⁻¹), indicating a rapid electron transfer rate and superior reaction kinetics (Fig. 3b and Supplementary Fig. 13). Regarding the UOR potential, the asymmetric Ni–O–Ti sites outperformed most of the previously reported Ni-based electrocatalysts (Supplementary Fig. 14 and Table 3).

Regardless of their comparable UOR potentials in catalyzing the UOR, asymmetric Ni–O–Ti sites and symmetric Ni–O–Ni sites exhibited significant differences in their selectivity for N$_2$ evolution. Systematic analyses using ion chromatography and on-line mass spectrometry within a potential range of 1.40 to 1.70 $V_{RHE}$ (Supplementary Figs. 15–18) revealed that the asymmetric Ni–O–Ti sites primarily generated innocuous N$_2$ (1), achieving a N$_2$ selectivity exceeding 99% (Fig. 3c, Supplementary Figs. 19–21 and Supplementary Movie 1)[17,18,31,32]. Concurrently, a notable H$_2$ evolution rate of 22.0 mL h⁻¹ was delivered at the coupled Pt cathode under a current density of 213 mA cm⁻² at 1.40 $V_{RHE}$ (2–3). Symmetric Ni–O–Ni sites, however, exhibited a poor N$_2$ selectivity of 61% under 1.40 $V_{RHE}$, which decreased progressively to 56%, 14%, and 9% as the potential increased to 1.50, 1.60, and 1.70 $V_{RHE}$, respectively (Supplementary Figs. 22, 23 and Table 4), accompanied by a substantial production of NCO⁻ and overoxidized NO$_x^-$ (4). Moreover, conventional Ni foam displayed a disappointingly low N$_2$ selectivity ranging from 10% to 30%, illustrating that Ni–O–Ni sites disfavored N$_2$ production (Supplementary Figs. 24, 25). Furthermore, the indispensable contribution of Ti within the asymmetric Ni–O–Ti sites to the high N$_2$ selectivity of the UOR was confirmed by the poor N$_2$ selectivity (below 10%) of single Ni atoms dispersed on N-doped graphene

(Ni$_1$@NC) (Supplementary Fig. 26).

$$\text{Anode UOR} \qquad CO(NH_2)_2 + 6OH^- \rightarrow N_2 + CO_2 + 5H_2O + 6e^- \qquad (1)$$

$$\text{Cathode HER} \qquad 6H_2O + 6e^- \rightarrow 3H_2 + 6OH^- \qquad (2)$$

$$\text{Overall} \qquad CO(NH_2)_2 + H_2O \rightarrow N_2 + 3H_2 + CO_2 \qquad (3)$$

$$\text{Overoxidation} \qquad CO(NH_2)_2 + 8OH^- \rightarrow NCO^- + NO_2^- + 6H_2O + 6e^- \qquad (4)$$

We subsequently utilized online mass spectrometry to analyze the mass-to-charge ratio of the produced N$_2$ during UOR by feeding CO($^{14}$NH$_2$)$_2$ and CO($^{15}$NH$_2$)$_2$ in a 1:1 ratio. The observed mass spectral distribution of $^{14}$N$_2$:$^{15}$N$_2$:$^{14}$N$^{15}$N at a 1:1:0 ratio offered compelling evidence of a dominant intramolecular N–N bond coupling reaction mediated by the asymmetric Ni–O–Ti sites (Fig. 3d)[24]. Control experiments indicated that the introduction of external 0.1 M NCO⁻ and 0.1 M NO$_2^-$ in the electrolyte led to the poisoning of asymmetric Ni–O–Ti sites, as evidenced by decreased current densities (Fig. 3e). To this end, the prevention of secondary NCO⁻ and NO$_x^-$ generation over the asymmetric Ni–O–Ti sites ensured the stability of the electrochemical system over 10,000 cycles and an extended period of 10 days with a well-maintained N$_2$ selectivity of 99% during the laboratory test (Fig. 3f and Supplementary Figs. 27–30). A minor reduction in the current density was attributed to the depletion of urea as the UOR proceeded, which could be easily solved by refreshing the electrolyte. In contrast, the Ni–O–Ni sites and Ni foam exhibited more pronounced decreases in current density, retaining only 64.1% and 16.4% of their initial values, respectively.

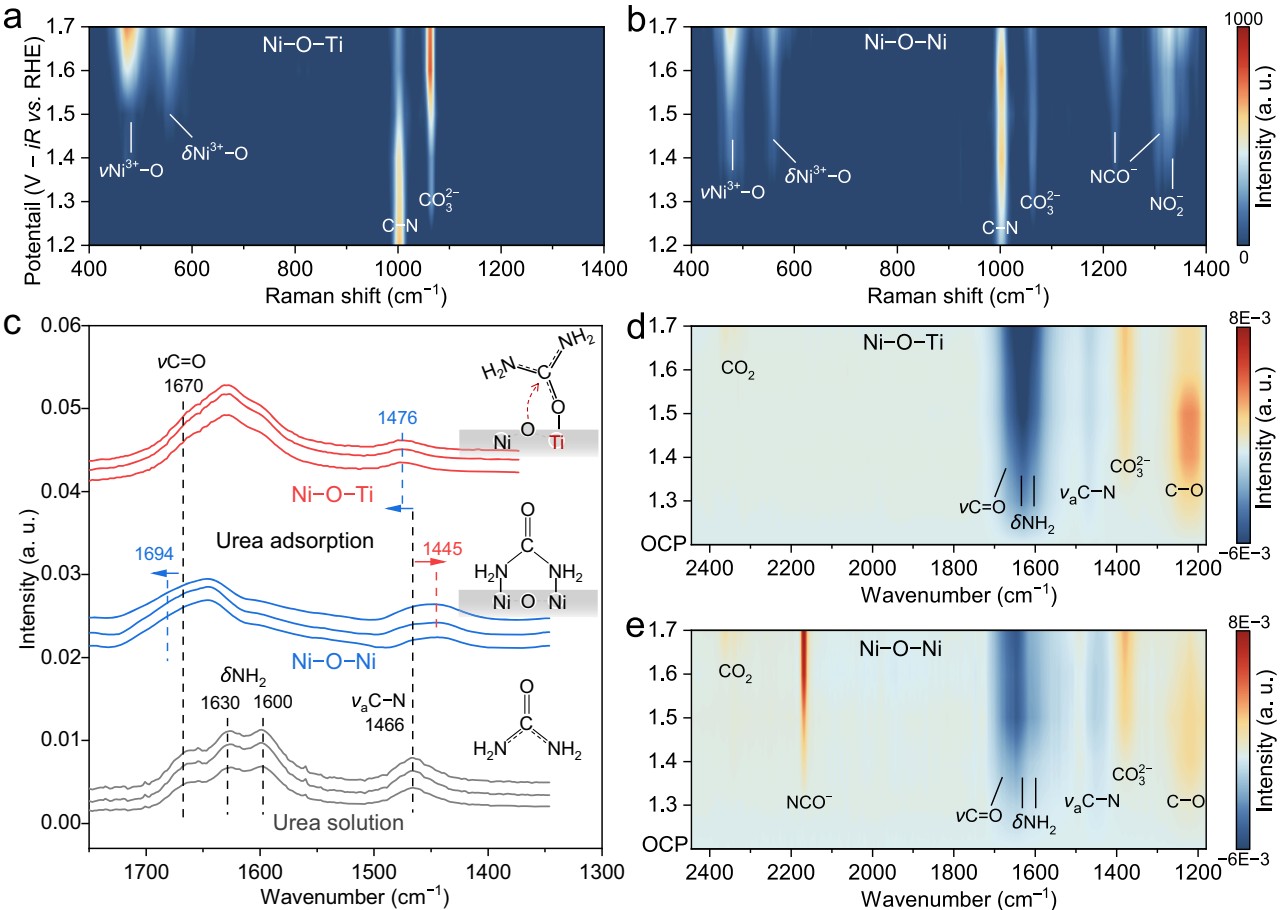

**Fig. 4 | Mechanism of the UOR on the atomically isolated asymmetric Ni–O–Ti sites.** In situ Raman spectra of (**a**) atomically isolated asymmetric Ni–O–Ti sites and (**b**) connected symmetric Ni–O–Ni sites at different potentials during the UOR operations. The distinct Raman peaks at 476 and 556 cm$^{-1}$ are attributed to $E_g$ bending vibration and the $A_{1g}$ stretching modes of Ni$^{3+}$–O, respectively. **c** The FTIR spectra of urea solution and adsorbed urea on atomically isolated asymmetric Ni–O–Ti sites and connected symmetric Ni–O–Ni sites. In situ FTIR spectra during the UOR on (**d**) atomically isolated asymmetric Ni–O–Ti sites and (**e**) connected symmetric Ni–O–Ni sites at different potentials.

## Reaction mechanism of the UOR

In situ Raman spectroscopy was utilized to clarify the role of atomically isolated asymmetric Ni–O–Ti sites in the selective oxidation of urea to N$_2$ (Supplementary Fig. 31)[33]. The UOR on Ni-based electrocatalysts is known to follow a sequential electrochemical-chemical mechanism. That is, Ni first undergoes oxidation into high-valent Ni$^{3+}$–O species. Subsequently, Ni$^{3+}$–O reacts with the adsorbed urea to produce CO$_2$, N$_2$, and H$_2$O, simultaneously regenerating the catalysts. This process was confirmed by the appearance of Ni$^{3+}$–O on the asymmetric Ni–O–Ti sites at potentials ≥1.30 V$_{RHE}$ (Supplementary Fig. 32)[34,35]. A slightly greater potential was required for the formation of Ni$^{3+}$–O species on the symmetric Ni–O–Ni sites and on the Ni foam (1.40 V$_{RHE}$), which is consistent with the LSV curves of these materials. Upon the introduction of urea with a prototypical C═N stretching band at 1004 cm$^{-1}$ (Fig. 4a)[12], the Ni$^{3+}$–O Raman peak rapidly diminished and completely vanished by 1.50 V$_{RHE}$ along with the emergence of adsorbed CO$_2$ species (CO$_3^{2-}$) at 1065 cm$^{-1}$, highlighting Ni$^{3+}$–O as the key species for the UOR (Supplementary Fig. 33)[36]. Interestingly, neither NCO$^-$ nor NO$_x^-$ was observed on the asymmetric Ni–O–Ti sites (Supplementary Fig. 34), indicating that the C═N bond was well stabilized prior to intramolecular N–N coupling. In contrast, both the Ni–O–Ni sites and the Ni foam exhibited pronounced accumulations of NCO$^-$ (1221 and 1313 cm$^{-1}$) and NO$_2^-$ (1331 cm$^{-1}$) species during the UOR, reflecting their poor selectivity for N$_2$ evolution (Fig. 4b and Supplementary Fig. 35).

To clarify the specific reaction pathways governed by the asymmetric Ni–O–Ti sites for the UOR, an in situ FTIR spectroscopy study

was conducted[37]. Urea features $\nu$(C═O) at 1670 cm$^{-1}$, $\delta$(NH$_2$) at 1630 and 1600 cm$^{-1}$, and $\nu_a$(C═N) at 1466 cm$^{-1}$ (Fig. 4c and Supplementary Fig. 36)[38]. As mentioned above, the C═N bond in urea possesses 30% hybridized double bonding characteristics due to the resonance between the NH$_2$ and C═O groups. Upon interaction with symmetric Ni–O–Ni sites via NH$_2$ groups, the C═N bond was weakened due to increased electron demand from donor N atoms, disrupting the resonance between the NH$_2$ and C═O groups. This interaction is demonstrated by a redshift of $\nu_a$(C═N) to 1445 cm$^{-1}$ and a blueshift of $\nu$(C═O) to 1694 cm$^{-1}$, indicative of the weakening of the C═N bond and the strengthening of the C═O bond, respectively. These spectral shifts, in conjunction with the DFT simulations, corroborate the dual NH$_2$ adsorption configuration of urea at symmetric Ni–O–Ni sites (Supplementary Fig. 37). In contrast, a blueshift of $\nu_a$(C═N) to 1476 cm$^{-1}$ was observed for urea adsorbed on the Ni–O–Ti sites, suggesting a strengthening of the C═N bond (Supplementary Fig. 38). Benefiting from the high oxygenophilicity of Ti, urea might bind to Ni–O–Ti sites in a new adsorption configuration via C═O→Ti interactions. With an increase in the potential to 1.30 V$_{RHE}$, the IR peak assigned to urea gradually diminished, while a distinct peak attributed to the stretching vibration of the C–O single bond emerged at 1250 cm$^{-1}$ (Fig. 4d)[9,39]. We believe that this peak is derived from the adsorption of the C═O group on Ni$^{3+}$–O in the Ni–O–C═O→Ti configuration[40]. The peak at ~1380 cm$^{-1}$ arose from CO$_3^{2-}$ in the electrolyte and gradually accumulated with increasing potential along with the physically adsorbed CO$_2$ (2343 cm$^{-1}$)[41-43]. The absence of NCO$^-$ or

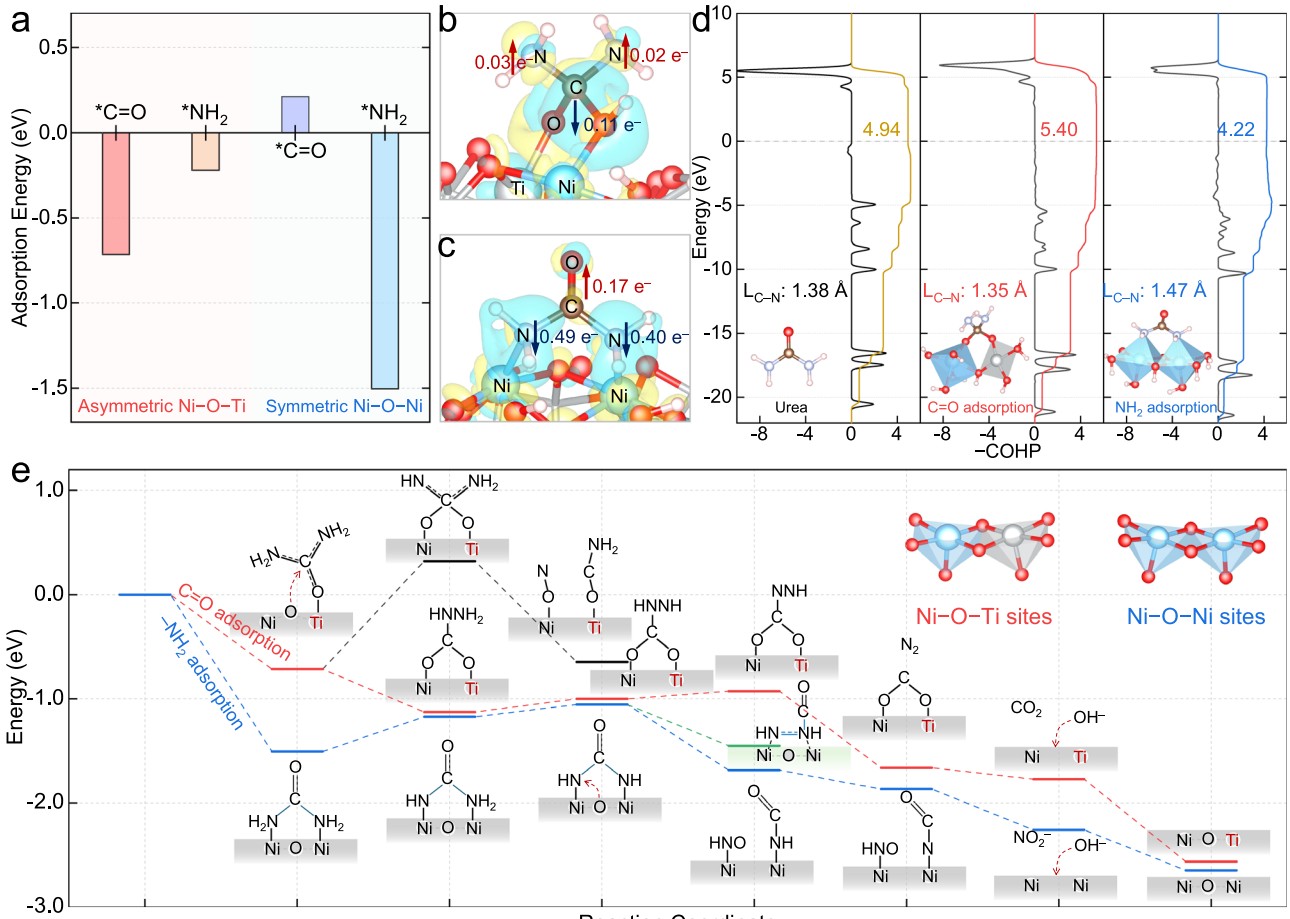

**Fig. 5 | Theoretical investigation of the UOR on the atomically isolated asymmetric Ni–O–Ti sites. a** The adsorption energies of urea on asymmetric Ni–O–Ti sites and symmetric Ni–O–Ni sites in different configurations. Charge density difference of urea adsorbed on (**b**) asymmetric Ni–O–Ti sites and (**c**) symmetric Ni–O–Ni sites. The yellow and blue iso-surfaces with an iso-value of 0.004 a.u. represent spatial charge accumulation and depletion, respectively. **d** The COHP analysis of C–N bond in the free and adsorbed urea on asymmetric Ni–O–Ti and symmetric Ni–O–Ni sites. **e** The UOR process diagram and Gibbs free energy change on asymmetric Ni–O–Ti and symmetric Ni–O–Ni sites.

$NO_x^-$ vibrational peaks in the spectra of asymmetric Ni–O–Ti sites across all potentials further confirms their high $N_2$ selectivity. In contrast, symmetric Ni–O–Ni sites displayed an increasing NCO⁻ peak at 2168 cm⁻¹, associated with robust C═N bond cleavage of urea (Fig. 4e)[44].

**Theoretical investigation of the UOR**

Building on the experimental results and discussions, comprehensive density functional theory (DFT) calculations were conducted to clarify the UOR pathways governed by asymmetric Ni–O–Ti sites (Supplementary Fig. 39 and Supplementary Data 1). As previously demonstrated, the Ni–O–Ti sites favored interactions with the C = O group of urea with higher adsorption energy (− 0.71 eV) than did interactions with the $NH_2$ groups (− 0.22 eV). The electrochemically generated $Ni^{3+}$–O then bound to the C moiety of the C = O group (Supplementary Fig. 40), while the oxygenophilic Ti atom was associated with the O moiety, forming an asymmetric bidentate binuclear intermediate (Fig. 5a). Significantly, this urea adsorption configuration on Ni–O–Ti sites manifested an electron accumulation on the two C═N bonds (+ 0.02 e and + 0.03 e), indicating a strengthened resonance between the $NH_2$ and C = O groups (Fig. 5b). This was in stark contrast to the Ni–O–Ni sites where urea adsorption through its $NH_2$ groups with a larger adsorption energy of − 1.50 eV led to a decrease in the electron density on the C═N bonds (− 0.40 e and − 0.49 e, Fig. 5c). The strengthened resonance within the urea molecule further shortened

the C═N bond length from 1.38 Å to 1.35 Å and increased the bond strength from 4.94 eV to 5.40 eV, according to crystal orbital Hamilton population (COHP) analysis (Fig. 5d)[45–48]. In contrast, the resonant C═N bond was elongated to a single C–N bond of 1.47 Å with a destabilized bond strength of 4.22 eV[49]. Thus, the asymmetric Ni–O–Ti sites might prevent premature breakage of the primary C═N bond prior to intramolecular N–N coupling.

The elemental steps of the UOR were then explored through broader theoretical calculations (Fig. 5e and Supplementary Figs. 41–43). Intriguingly, the urea adsorbed on the asymmetric Ni–O–Ti sites spontaneously underwent intramolecular N–N coupling during the initial dehydrogenation step, forming an *OCNH–$NH_2$ intermediate while releasing 0.41 eV of energy. This pathway was more thermodynamically favorable than the alternative route, which involves the breaking of the C═N bond to form *ON + *OCNH₂ intermediates, a reaction confronted with a large potential-independent energy barrier of + 1.03 eV. Subsequent stepwise dehydrogenation and desorption steps led to the proactive production of $N_2$ with a relatively minor energy barrier of + 0.13 eV, consistent with the high UOR selectivity toward $N_2$ on the asymmetric Ni–O–Ti sites. In contrast, at the symmetric Ni–O–Ni sites, urea necessitated a higher energy input (+ 0.34 eV) for the first dehydrogenation step to generate *NHCONH₂. In the subsequent dehydrogenation stage, the generated *NHCONH was more inclined to undergo C═N bond cleavage into *HNO + *HNCO (Supplementary Fig. 44), a route that is more thermodynamically

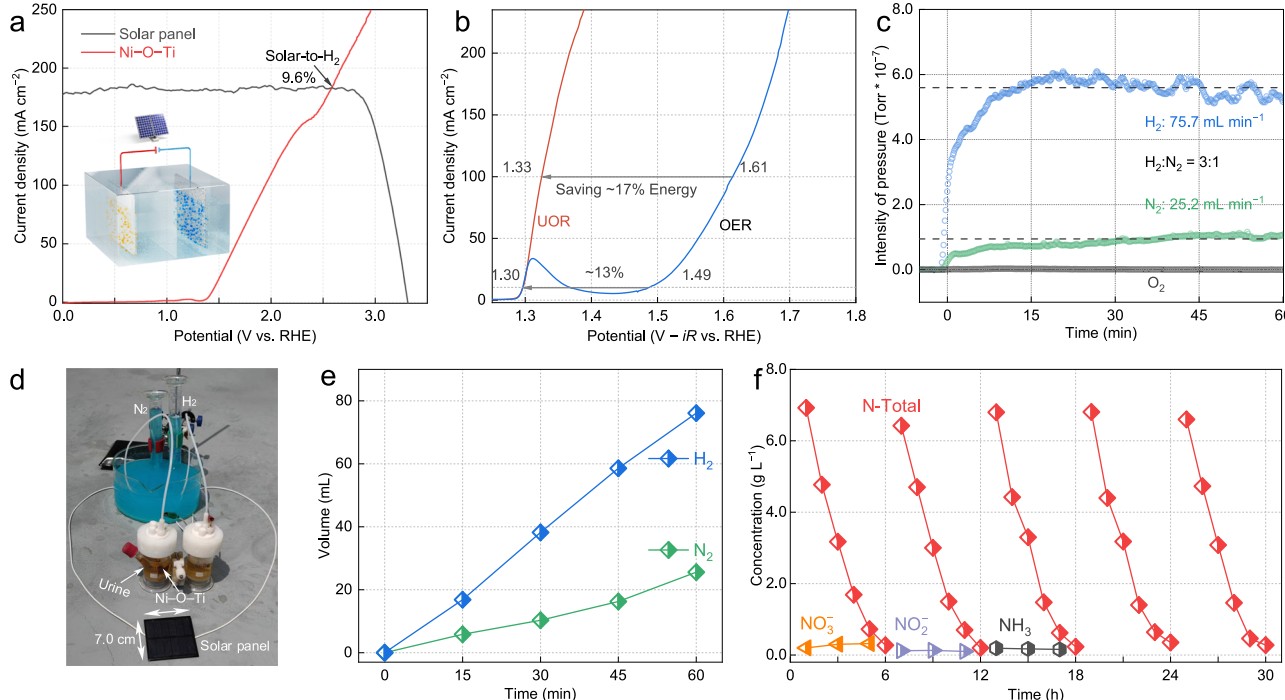

**Fig. 6 | Solar-powered reactor design and performance. a** J–V curve of the solar panel and the UOR polarization curves with the Ti foam containing atomically isolated asymmetric Ni–O–Ti sites as the anode and Pt foil as the cathode with a scan rate of 5 mV s$^{-1}$. **b** LSV curves with 85% *iR* correction of the asymmetric Ni–O–Ti sites in 1.0 M KOH + 0.33 M urea and 1.0 M KOH for driving the UOR and OER, respectively. The on-line MS of $H_2$ and $N_2/O_2$ collected from cathode and anode chambers, respectively, (**c**) in 1.0 M KOH + actual urine (urea: ~ 7 g L$^{-1}$). **d** The digital image of the solar-powered reactor. **e** The collected $H_2$ and $N_2$ from actual urine. **f** Concentrations of N-total, $NO_3^-$, $NO_2^-$ and $NH_3$ during photo-electrochemical urine oxidation process.

favorable by 0.23 eV over the formation of the *HN–NHCO intermediate via intramolecular N–N coupling, thus accounting for the excessive and uncontrolled oxidation of urea into $NO_2^-$ and $NCO^-$ on conventional Ni-based catalysts during the UOR.

## Solar-powered UOR and H2 evolution
To showcase the practical application of these asymmetric Ni–O–Ti sites, we designed a prototype UOR–HER device powered by a commercial monocrystalline Si solar panel (3.0 V, 49 cm$^2$)[50]. The projected operation current of the designed system was estimated to be 183 mA according to the intersection of the J–V curve of the solar cell and the polarization curve of the electrolysis device (Fig. 6a). Compared with the OER, the potential required to reach a current density of 100 mA cm$^{-2}$ on asymmetric Ni–O–Ti sites for the UOR was reduced by 0.28 V, resulting in an energy savings of ~17% for $H_2$ production (Fig. 6b, Supplementary Fig. 45 and Supplementary Notes 2). Solar-powered UOR–HER electrolysis was then conducted on a clear summer day in Shanghai, China, using actual urine collected from public facilities and commercial Pt foil as the cathode electrode, delivering $H_2$ and $N_2$ production rates of approximately 3:1, with 75.7 and 25.2 mL h$^{-1}$, respectively, and a solar-to-$H_2$ conversion efficiency of 9.6% (Fig. 6c–e and Supplementary Fig. 46). The Faradaic efficiency of urea-to-$N_2$ conversion and $H_2$ production in this solar-powered reactor exceeded 99%. After 6 h of the electrochemical reaction, 96% of the N total in the urine was removed (Fig. 6f). In addition, the concentrations of $NO_3^-$, $NO_2^-$, and $NH_4^+$ were maintained at low levels throughout the other 4 cycles, revealing the high efficiency and stability of asymmetric Ni–O–Ti sites for practical UOR applications.

## Discussion
We have demonstrated the fabrication of atomically isolated asymmetric Ni–O–Ti sites on a Ti foam anode with an $N_2$ selectivity of 99% for the UOR, surpassing the connected symmetric Ni–O–Ni counterparts in documented Ni-based electrocatalysts with $N_2$ selectivity below 55% and delivering a $H_2$ evolution rate of 22.0 mL h$^{-1}$ when coupled to a Pt counter cathode under 213 mA cm$^{-2}$ (1.40 $V_{RHE}$). The oxygenophilic Ti enables urea adsorption on the asymmetric Ni–O–Ti sites via C = O, reinforcing the resonance between the $NH_2$ and C = O groups to avoid premature breakage of the C≡N bond and favor selective intramolecular N–N coupling to $N_2$. Moreover, the prevention of poisoning by secondary $NCO^-$ and $NO_x^-$ ensures the stability of the electrochemical system over an extended duration of 10 days. We have also conceptualized and engineered a prototype solar-powered UOR–HER device for on-site urine processing, which offers an eco-friendly solution to process urine in underdeveloped areas with limited infrastructure, reducing nitrogen pollution while supporting decentralized clean energy production.

## Methods
### Chemicals and materials
The Ti foam electrode (0.68 mm thick, 99.99%) utilized in the study was procured from Kunshan Guangjiayuan New Material Co., Ltd., China. Ni foam (1.0 mm thick, Tanqi New Materials Technology Department, Shanghai) and Pt foil (0.1 mm thick, 99.99%, GaossUnion) were used as received. The chemical reagents, nickel(II) chloride hexahydrate ($NiCl_2 \cdot 6H_2O$, AR), potassium hydroxide (KOH, GR, 95%), potassium nitrate ($KNO_3$, AR, ≥ 99.5%), potassium nitrite ($KNO_2$, AR, 97%), potassium cyanate (KOCN, AR, 97%), heavy-oxygen water ($H_2^{18}O$, 97atom% $^{18}O$), urea (CO($^{14}NH_2$)$_2$, 99.5%) and $^{15}$N-labeled urea (CO($^{15}NH_2$)$_2$, AR, 99atom%) and ethanol were sourced from Sinopharm Chemical Regent Company. All chemicals were used without further purification. Ultrapure water (R = 18.25 mΩ) has been applied in all the experiments.

### Electrodes
The atomically isolated asymmetric Ni–O–Ti sites on the Ti foam were prepared using the droplet coating and annealing method. Typically,

50.0 μL, 100.0 μL and 200.0 μL homogeneous $NiCl_2 \cdot 6H_2O$ ethanol solution with a Ni mass concentration of $3.0\ mg\ mL^{-1}$ was uniformly deposited onto $1.0\ cm \times 0.5\ cm$ Ti foam, respectively. The deposition process was expedited by infrared lamp illumination, which accelerated solvent evaporation and prevented solvent aggregation arising from surface tension. Subsequently, the Ni-deposited electrodes were treated in a reductive atmosphere ($5\%\ H_2/Ar$) at 300 °C for 3 hours with a heating rate of $10\ °C\ min^{-1}$ to yield the atomically isolated asymmetric Ni–O–Ti sites. The counterpart connected symmetric Ni–O–Ni sites were synthesized by increasing the volume of the homogeneous $NiCl_2 \cdot 6H_2O$ solution to 800.0 μL (Supplementary Table 1). The $Ni_1@NC$ was synthesized by high-temperature calcination (Supplementary Notes 3).

## Characterization

XRD patterns were collected using a Rigaku Miniflex-600 model with Cu Kα radiation ($\lambda = 0.15406\ nm$), operated at 40 kV and 15 mA. HAADF-STEM images were acquired utilizing a JEOL JEM-ARM200F TEM/STEM, equipped with a spherical aberration corrector and operated at 200 kV. ICP-MS analyses were performed on a Thermo Scientific iCAP Q. XPS assessments were performed with a PHI 5000 Verasa scanning X-ray microprobe from ULAC-PHI, Inc., using Al Kα radiation. The binding energy calibration was standardized to the C $1s$ peak at 284.6 eV. STEM mapping was obtained on an FEI Talos F200X. The X-ray absorption fine structure spectra of Ni K-edge were collected at the Singapore Synchrotron Light Source Center (Supplementary Notes 4). In situ Raman spectra were acquired using a Renishaw inVia Qontor instrument, employing a 532 nm semiconductor laser as the illumination source. The FT-IR spectra were recorded on a Nicolet iS50FT-IR spectrometer (Thermo, USA), scanning from 4000 to $800\ cm^{-1}$ at room temperature. A chronoamperometry test was performed for 600 s to collect FT-IR and Raman spectra. Spin-polarized DFT calculations were conducted using the Vienna Ab initio Simulation Package (Supplementary Notes 5 and Supplementary Data 1)[51,52].

## Electrochemical measurements

Electrochemical data were collected in a conventional three-electrode system operated by a CHI 760E electrochemical analyzer (Shanghai Chenhua, China) at ambient conditions ($25 \pm 2\ °C$). The UOR electrodes, Pt foil, and Hg/HgO electrodes were used as working electrodes, counter electrodes, and reference electrodes, respectively. All of the electrodes were assembled within a H-type cell. The working and counter electrode was spaced $2.0 \pm 0.5\ cm$ and separated by a Nafion 117 membrane ($4.5 \times 4.5\ cm^2$) with a thickness of 183 μm. The membrane underwent an initial treatment in 5 wt% $H_2O_2$ at 80 °C for an hour, before a half-hour soak in deionized water. Subsequently, it was boiled in 5 wt% $H_2SO_4$ at the same temperature for an hour. Finally, it was soaked in deionized water for half-hour. The immersed area of the working electrode was $0.25\ cm^2$ ($0.5 \times 0.5\ cm^2$) in 20.0 mL electrolyte. If not specified otherwise, all potentials reported in this work were referenced to an RHE (Supplementary Notes 6). Before each measurement, the electrolyte was purged with Ar (99.99%) for 30 min. LSV curves were recorded three times at a scan rate of $5\ mV\ s^{-1}$ in a 1.0 M KOH that contains 0.33 M urea with a pH value of $13.7 \pm 0.1$. The electrolyte in the cathodic compartment was stirred at a rate of 500 rpm during electrolysis. Chronopotentiometry tests for the UOR selectivity were conducted at certain applied potentials within the UOR window for 30 min. The durability tests were carried out using the chronoamperometry measurements under $1.33\ V_{RHE}$ with the initial current density of $100\ mA\ cm^{-2}$ in 1.0 M KOH that contains 1.0 M urea. The electrochemical impedance spectroscopy (EIS) measurements were carried out in the frequency range from 100 kHz to 0.1 Hz. The charge transfer resistance ($R_{ct}$) obtained by fitting the EIS data was used for the $iR$ compensation. The Faradaic efficiency (FE) was calculated on the basis of the following Eq. (5):

$$FE = (nF \times N)/Q \times 100\% \tag{5}$$

where $n$ is the number of electrons transferred for each product molecule. Specifically, the values of $n$ for $NCO^-$, $H_2$, $O_2$, $N_2$, and $NO_2^-$ are 0, 2, 4, 6 and 6, respectively. $F$ is Faraday's constant ($96,485\ C\ mol^{-1}$), $N$ is the mole number of products, and $Q$ is the number of charges passing through the electrode.

## Determination of urea and N-total

Urea was quantified by high-performance liquid chromatography (HPLC, Thermo Scientific™ UltiMate™ 3000) method using a mobile phase of acetonitrile-water (40:60, V/V) with a flow rate of $1.0\ mL\ min^{-1}$. The HPLC column was Agilent ZORBAX SB-C18 ($4.6 \times 150\ mm$). The detection wavelength was 195 nm. The injection volume was 10 μL. The N-total concentration was determined by using UV spectrophotometry according to the Chinese Standard HJ636–2012. Briefly, the nitrogen of nitrogen compounds was converted into nitrate first by digestion using alkaline potassium persulfate for 30 min at 120 °C. The absorbance was measured at wavelengths of 220 nm and 275 nm after acidification with concentrated HCl. The calibrated absorbance was determined by taking the absorbance at 220 nm and subtracting double the absorbance at 275 nm, showing a clear proportional relationship to the total nitrogen content.

## Solar-powered H2 production

A commercial monocrystalline Si solar panel was used for solar-driven $H_2$ production from urine. The anode of the solar reactor was fabricated by atomically isolated Ni–O–Ti sites on $2.0 \times 2.0\ cm^2$ commercial Ti foam. Commercial Pt foil ($1.0 \times 2.0\ cm^2$) was used as a cathode, separated by a Nafion 117 membrane. The solar-to-$H_2$ efficiency (SE) was calculated according to the following Eq. (6):

$$SE = (j_{UOR} \times E \times FE)/(A_{solarpanel} \times P^{in}) \times 100\% \tag{6}$$

where $j_{UOR}$ is the UOR current (183 mA). $E$ is the stable cell potential (2.6 V). FE is the Faradaic efficiency of HER (99%). The $A_{solarpanel}$ is the solar-panel area ($49.0\ cm^2$). The $P^{in}$ is the optical power ($100\ mW\ cm^{-2}$). The urine electrolysis measurement under natural sunlight was conducted on a clear summer day at the location of 31°1′29.69″ N 121°25′48.02″ E Shanghai, China.

## Data availability

The datasets analyzed and generated during the current study are included in the paper and its Supplementary Information. The data that support the findings of this study are available from the corresponding author upon request. Source data are provided with this paper.

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

## Acknowledgements

This work was supported by the National Natural Science Foundation of China (U22A20402, L.Z.), (22206124, H.L.), (22102100, Y.Y.), (22306119, Y.S.), the National Key Research and Development Program of China (2022YFA1505000, H.L.), (2021YFA1201701, Y.Y.), the Natural Science Foundation of Shanghai (23ZR1431000, H.L.), (22ZR1431700, Y.Y.), Shenzhen Science and Technology Program (JCYJ20220818095601002, Y.Y.), the China Postdoctoral Science Foundation (2022M722080, G.Z.), (2021M702117, J.D.), (2022M712049, 2023T160419, Y.S.), and the Shanghai Postdoctoral Excellence Program (2022381, G.Z.), (2021182, J.D.). The authors acknowledge the support from the Instrumental Analysis Center of Shanghai Jiao Tong University, the Instrumental Analysis Center of the School of Environmental Science and Engineering, Shiyanjia Lab (www.shiyanjia.com), and ceshihui (www.ceshihui.cn). The computations were run on the π 2.0 cluster supported by the Center for High-Performance Computing at Shanghai Jiao Tong University.

## Author contributions

L. Z. Z., H. L., and Y. C. Y. supervised the project. G. M. Z., H. L., and Y. C. Y. conceived and designed the experiments. G. M. Z. conducted material synthesis and characterizations, electrochemical experiments, and DFT calculations. L. F. H. contributed to the electrochemical experiments. L. Z. contributed to the material characterizations. J. D., Q. Z., X. Y. Z., Y. B. S., J. X. W., and W. H. commented on the paper. G. M. Z., H. L., and L. Z. Z. wrote the paper.

## Competing interests

The authors declare no competing interests.
