## [Peer Review File · Nature Communications]

REVIEWER COMMENTS

Reviewer #1 (Remarks to the Author):

This work demonstrates that isolated asymmetric Ni–O–Ti sites on a Ti foam anode achieve a remarkable N₂ selectivity of 99% for UOR. In general terms, this is a relevant field in electrocatalysis, as urea oxidation can be extremely useful in both energy-conversion technologies and energy-saving fields. The authors proposed a well-planned set of experiments, supported by computational modeling and In-situ characterization (Raman, FTIR, and MS), to elucidate the mechanism of urea dissociation at Ni–O–Ti active sites with high N₂ selectivity. This study lacks novelty. There are some aspects that the authors could improve or clarify the paper.

1. The comparison of M-O bond energies should focus on similar space point groups and identical coordination environments (lines 60-65). The author needs to specify which materials' M-O bond energies were compared in this context.
2. Some data are overinterpreted in the in-situ FTIR spectroscopy section (lines 191-196). For instance, attributing the redshift of $\nu(\text{C}=\text{N})$ and blueshift of $\nu(\text{C}=\text{O})$ to the dual Ni-N adsorption configuration, and the blueshift of $\nu(\text{C}=\text{N})$ to the strengthening of the C-N bond, it is suggested that the author provide further explanations.
3. No evidence of lattice oxygen participation was found in the main text. In DFT calculations of the urea dissociation process at different sites, why did the authors choose lattice oxygen as the source of HNO* / CO₂ oxygen instead of solution OH-.
4. Supplementary Figure 26. It is recommended that the author provide information on AIMD convergence accuracy, simulation temperature, ensemble, etc.
5. The author needs to check for other typo errors, such as in Supplementary Figure 17 line 127, where it should be "1M KOH" instead of "1M KCl".

Reviewer #2 (Remarks to the Author):

In this manuscript, the authors address the critical issue of poor N₂ selectivity in the urea oxidation reaction (UOR) by constructing atomically isolated asymmetric Ni-O-Ti sites on commercial Ti foam. The asymmetric sites alter the urea adsorption configuration, thereby preventing C=N cleavage and significantly enhancing N₂ selectivity. The resulting anode with Ni-O-Ti sites demonstrates robust performance in actual urine denitrification and is utilized in a UOR-HER prototype for efficient solar-to-H₂ conversion. The specific comments are as follows:

1. From a theoretical perspective, why does the Ti foam containing symmetric Ti-O-Ti appear relatively inert for UOR?

2. The electrochemical surface area (ECSA) is crucial in the electrocatalytic process and should be measured, as it may influence the UOR activities of Ni-O-Ni and Ni-O-Ti sites.
3. The potential dissolution of Ni from the Ti foam into the aqueous solution during UOR, due to oxidation of Ni to a high valence state, should be investigated.
4. The stability of the Ti foam with Ni single atoms after a long-term electrochemical reaction should also be examined.
5. The use of different urea concentrations in linear sweep voltammetry (0.33 M) and stability tests (1.0 M) raises questions about the activity of the Ni-O-Ti sites under varying urea concentrations.
6. The methods for detecting urea and total nitrogen concentrations should be specified in the Methods section of the main text.
7. The authors should provide the k-space of the extended X-ray absorption fine structure (EXAFS) fitting results for both asymmetric Ni-O-Ti and symmetric Ni-O-Ni sites to show the reasonableness of the fitting results.
8. As shown in Fig. 5b, what is the calculation method for energy-saving efficiency in H₂ production?

Reviewer #3 (Remarks to the Author):

In this manuscript, Zhan et al. report an interesting single-atom electrochemical approach for urea pollutant control and resource utilization based on high selective urea oxidation reaction (UOR) coupled hydrogen evolution reaction (HER). They found that atomically isolated asymmetric Ni-O-Ti sites could prevent the generation of harmful byproducts (CNO⁻ or NO_x⁻) and delivered an impressive N₂ selectivity of 99% during UOR with improved stability. The conceptualization and engineering of a prototype UOR-HER device, powered by commercial silicon photovoltaic cells, highlights an avenue for urea control and decentralized hydrogen production. Therefore, I recommend its publication in Nature Communications after some minor revisions. Specific comments are as follows.

1. How did the strong C=O→Ti interaction contribute to urea oxidation and N₂ evolution? Could urea be adsorbed on freestanding Ti foam?
2. Please provide the electronic state of Ni single atoms after UOR to assess the stability of the electrode.
3. Does the preparation temperature influence the UOR performance of Ni-O-Ti sites?
4. The specific contribution of Ni-O-Ti shells in FT-EXAFS R-space should be determined to support Ni-O-Ti coordination more clearly.
5. The charge transfer resistance of Ti foams with Ni-O-Ti and Ni-O-Ni sites may affect their UOR activity, which should be measured using electrochemical impedance spectroscopy.
6. Were Nafion membranes applied to treat actual urine in the UOR-HER device? The utilization of Nafion membranes may increase the system resistance.
7. Some very recent literatures about UOR should be cited, such as *Adv. Funct. Mater.*, 2023, 2303986; *J. Am. Chem. Soc.*, 2022, 144, 1174-1186; *Chem Catalysis* 2024, 4, 100840.

Point-by-point Response to Referees

Our responses and important changes in the revised manuscript and Supporting Information were all highlighted in BLUE.

Reviewer #1: This work demonstrates that isolated asymmetric Ni–O–Ti sites on a Ti foam anode achieve a remarkable N₂ selectivity of 99% for UOR. In general terms, this is a relevant field in electrocatalysis, as urea oxidation can be extremely useful in both energy-conversion technologies and energy-saving fields. The authors proposed a well-planned set of experiments, supported by computational modeling and *in-situ* characterization (Raman, FTIR, and MS), to elucidate the mechanism of urea dissociation at Ni–O–Ti active sites with high N₂ selectivity. This study lacks novelty. There are some aspects that the authors could improve or clarify the paper.

Response: Thank you for your insightful comments and constructive suggestions. Traditional Ni-based electrocatalysts for urea oxidation reaction (UOR) often suffer from poor selectivity towards innocuous N₂, thus producing detrimental NCO[−] or NO_x[−] byproducts that increase environmental risks. The development of UOR with high N₂ selectivity is highly desired but remains a great challenge.

Recent studies, such as those by Yang et al. (*Angew. Chem. Int. Ed.* 2021, **60**, 26656–26662), have made progress by improving the Faradaic efficiency of N₂ production from approximately 15.0% to 31.1% using a polyaniline-coated Ni foam. Similarly, Klinkova et al. (*Angew. Chem. Int. Ed.* 2022, **61**, e202209839) increased the Faradaic efficiency of N₂ production from 30% to 55% by doping a Ni(OH)₂/NF anode with copper to construct Ni_{0.8}Cu_{0.2}(OH)₂/NF. However, the N₂ selectivities of UOR in these pioneering studies were still not high.

In this study, we report the first example of UOR with a N₂ selectivity reaching up to 99% by constructing atomically isolated asymmetric Ni–O–Ti sites on a Ti foam anode, thus avoiding the

generation of harmful NCO^- or NO_x^- . The novelty of this study lies in the combination of Ni single atoms with high oxygenophilic Ti atoms to prevent the formation of symmetrical Ni–O–Ni sites, thereby altering the urea adsorption configuration from dual NH_2 on Ni–O–Ni sites to C=O ends on Ni–O–Ti sites (Fig. R1a). This O-terminal adsorption configuration of urea increased the electron density of oxygen atom and also enhanced the resonance between the NH_2 and C=O groups, which could amplify the double bond character and bond strength of the C=N bond, inhibiting the urea cleavage into NCO^- and NH_x , and thus significantly boosting the N_2 selectivity of urea oxidation (Fig. R1b).

Moreover, we have also engineered a prototype solar-powered UOR–HER device for on-site urine processing. This device offers an eco-friendly solution to process urine in underdeveloped areas with limited infrastructure, reducing nitrogen pollution while supporting decentralized clean energy production.

Figure R1. (a) The adsorption configurations of urea on symmetric Ni–O–Ni sites and asymmetric Ni–O–Ti sites. (b) UOR process diagram and Gibbs free energy change on symmetric Ni–O–Ni sites and asymmetric Ni–O–Ti sites.

1. The comparison of M–O bond energies should focus on similar space point groups and identical coordination environments (lines 60–65). The author needs to specify which materials' M–O bond energies were compared in this context.

Response: Thank you for your valuable comments. We apologize for any confusion caused by our previous description of the M–O bond energy. In this study, the M–O bond energy is defined as the standard enthalpy change for the formation of a diatomic species between O and the transition metal at 298.15 K through the following fission: $M-O \rightarrow M + O$, typically measured using spectroscopy or mass spectrometry (Haynes, W. M., CRC Handbook of Chemistry and Physics. CRC Press: Boca Raton, FL, **2014**). This well-defined M–O bond energy represents the oxygenophilic affinity of transition metals in general.

Regarding your specific concern about comparing M–O bond energies within similar space point groups and identical coordination environments, our study broadly compares M–O bond energies across different transition metals to identify general trends applicable to a wide range of metals, rather than focusing on variations specific to particular spatial or electronic configurations. This approach helps us understand fundamental, consistent properties of metal-oxygen interactions that are relevant across various metal types, which is essential for developing a generalized understanding in this field (*Inorg. Chem.* **2016**, 55, 9461–947). According to the given definition and careful assessment of the periodic table, the selection of Ti with high oxygenophilic affinity indeed altered the urea adsorption configuration from dual NH_2 on Ni–O–Ni sites to C=O ends on Ni–O–Ti sites based on the experimental results, thus improving the N_2 selectivity of urea oxidation.

During the revision, we added the definition of the M–O bond energies to the caption of Scheme 1 as follows: “(d) The bond formation energies of O with Ti, Ni, and other common transition metals

at 298.15 K, defined as the standard enthalpy change for the formation of a diatomic species between O and the transition metal”.

2. Some data are overinterpreted in the *in-situ* FTIR spectroscopy section (lines 191–196). For instance, attributing the redshift of $\nu_a(\text{C}=\text{N})$ and blueshift of $\nu(\text{C}=\text{O})$ to the dual Ni–N adsorption configuration, and the blueshift of $\nu_a(\text{C}=\text{N})$ to the strengthening of the C–N bond, it is suggested that the author provide further explanations.

Response: Thank you for your valuable comments. To further support the conclusions drawn from the *in-situ* FTIR spectroscopy of urea adsorption, we conducted DFT calculations to investigate the frequencies of $\nu_a(\text{C}=\text{N})$ and $\nu(\text{C}=\text{O})$ for urea molecule adsorbed in different configurations using the Gaussian 09 package (Supplementary Fig. 34). The computational models were optimized using open-shell UB3LYP method with the 6-31G* basis set (*J. Phys. Chem.* **1994**, 98, 11623–11627). SMD implicit solvent model were carried out to mimic the influence of water (*J. Phys. Chem. B* **2009**, 113, 6378–6396). The frequencies of adsorbed urea were scaled by factor of 0.9614 (*J. Phys. Chem.* **1996**, 100, 16502–16513).

For the dual NH_2 adsorption model on Ni–O–Ni sites, our DFT calculations revealed a redshift in $\nu_a(\text{C}=\text{N})$ due to the weakening of the C=N bond, and a blueshift in $\nu(\text{C}=\text{O})$ resulting from the strengthening of the C=O bond. These computational findings were in alignment with the shifts we observed in the *in-situ* FTIR spectra and were consistent with previously reported data by Aldaz et al. (*Langmuir* **1997**, 13, 2380–2389). This model supported the dual NH_2 adsorption hypothesis at Ni–O–Ni sites. Conversely, for the terminal O adsorption on Ni–O–Ni sites, our simulations indicated a strengthening of the C=N bond and a weakening of the C=O bond, leading to a blueshift in $\nu_a(\text{C}=\text{N})$ and a redshift in $\nu(\text{C}=\text{O})$, respectively. These shifts differed from those observed at Ni–O–Ni sites and

corroborated with experimental findings from Naguib et al. (*J. Am. Chem. Soc.* **2018**, 140, 10305–10314). These additional DFT results provided a solid scientific basis for our interpretations of the spectral changes observed in the *in-situ* FTIR studies.

During the revision, Supplementary Fig. 34 and the related discussion were added on page 35 in the revised supporting information. We modified the description of FTIR spectra on page 11 in revised manuscript as follows: “Upon interaction with symmetric Ni–O–Ni sites via NH₂ groups, the C=N bond was weakened due to increased electron demand from donor N atoms, disrupting the resonance between NH₂ and C=O groups. This interaction is demonstrated by a red-shift of $\nu_a(\text{C=N})$ to 1445 cm⁻¹ and a blue-shift of $\nu(\text{C=O})$ to 1694 cm⁻¹, indicative of the weakening of the C=N bond and the strengthening of the C=O bond, respectively. These spectral shifts, alongside DFT simulations, corroborate the dual NH₂ adsorption configuration of urea at symmetric Ni–O–Ni sites (Supplementary Fig. 34).”

Supplementary Figure 34. The experimental FTIR frequencies of C=O and C=N of urea adsorbed at the Ni–O–Ni sites and theoretical frequencies of C=O and C=N of urea adsorbed at the Ni–O–Ni sites in different configurations.

3. No evidence of lattice oxygen participation was found in the main text. In DFT calculations of the urea dissociation process at different sites, why did the authors choose lattice oxygen as the source of HNO*/CO₂ oxygen instead of solution OH⁻.

Response: Thank you for your valuable comments. We employed on-line MS to check the participation of lattice O in the CO₂ generation during UOR in H₂¹⁸O (*Nat. Commun.* **2024**, *15*, 2501; *Nature Chem.* **2017**, *9*, 457–465).

Initially, we observed the MS signal of C¹⁶O₂ (m/z = 44) by feeding 1.0 M K¹⁸OH + 0.33 M C¹⁶O(NH₂)₂ in H₂¹⁸O (Supplementary Fig. 37a and 37b), suggesting the participation of lattice ¹⁶O from Ni–¹⁶O–Ti sites. As the UOR progressed, the signals of C¹⁶O¹⁸O (m/z = 46) gradually increased, indicating that lattice ¹⁶O atoms were gradually depleted and substituted by ¹⁸O from the H₂¹⁸O. After the lattice ¹⁶O atoms in Ni–¹⁶O–Ti were sufficiently replaced with ¹⁸O atoms, the resulting Ni–¹⁸O–Ti electrode was used for the electrooxidation of urea in H₂¹⁶O containing 1.0 M K¹⁶OH + 0.33 M C¹⁶O(NH₂)₂. The instant production of C¹⁶O¹⁸O during the UOR strongly validated the direct involvement of lattice ¹⁸O (Supplementary Fig. 37c and d). The lattice ¹⁸O atoms in Ni–¹⁸O–Ti were gradually substituted by ¹⁶O from solution, leading to the production of C¹⁶O₂.

Additionally, ion chromatograms were used to check the participation of lattice O in the production of NO₂⁻ during UOR in H₂¹⁸O. Only N¹⁶O₂⁻ was detected during the early stages of UOR of Ni–¹⁶O–Ni sites by feeding 1.0 M K¹⁸OH + 0.33 M urea (C¹⁶O(NH₂)₂) in H₂¹⁸O solution (Supplementary Fig. 41a). For the Ni–¹⁸O–Ni electrode, N¹⁸O¹⁶O⁻ was generated during the early stages of UOR by feeding 1.0 M K¹⁶OH + 0.33 M urea (C¹⁶O(NH₂)₂) in H₂¹⁶O (Supplementary Fig. 41b). These experimental findings, supported by our DFT simulations, clearly demonstrated that lattice oxygen participates actively in the UOR process.

During the revision, Supplementary Fig. 37, 41 and its related discussion were added on pages

38, 42 in the revised supporting information.

Supplementary Figure 37. (a) The on-line MS signals for $m/z = 44$ and 46 from anode chamber during UOR on Ni-¹⁶O-Ti sites with the initial current density of 100 mA cm^{-2} by feeding $1.0 \text{ M K}^{18}\text{OH} + 0.33 \text{ M urea (C}^{16}\text{O(NH}_2)_2\text{)}$ in H_2^{18}O , and (b) corresponding schematic diagram of the on-line MS measurement process. (c) The on-line MS signals for $m/z = 44$ and 46 on ¹⁸O labelled Ni-¹⁸O-Ti sites during UOR by feeding $1.0 \text{ M K}^{16}\text{OH} + 0.33 \text{ M urea (C}^{16}\text{O(NH}_2)_2\text{)}$ in H_2^{16}O , and (d) corresponding schematic diagram of the on-line MS measurement process.

Supplementary Figure 41. The original data of ion chromatograms after the initial 10 min of UOR process for connected symmetric (a) Ni–¹⁶O–Ni sites by feeding 1.0 M K¹⁸OH + 0.33 M urea in H₂¹⁸O solution and (b) ¹⁸O labelled Ni–¹⁸O–Ni sites by feeding 1.0 M K¹⁶OH + 0.33 M urea in H₂¹⁶O solution.

4. Supplementary Figure 26. It is recommended that the author provide information on AIMD convergence accuracy, simulation temperature, ensemble, etc.

Response: Thank you for your valuable comment. We are sorry for the missing information about AIMD. The AIMD simulation was performed using VASP, and the computational details were also reported in recent works (*J. Am. Chem. Soc.* **2024**, 146, 11152–11163; *Angew. Chem. Int. Ed.* **2022**, 61, e202208215; *Adv. Energy Mater.* **2021**, 11, 2002816; *Nat. Commun.* **2019**, 10, 5692.). A fixed volume canonical ensemble was applied to conduct anneal-to-quench process from 1500 K to 298 K with a series of AIMD simulations, and each experiment was conducted at a constant temperature using Nose–Hoover method (*J. Chem. Phys.* **1992**, 97, 2635–2643.). Initial temperature of 1500 K was chosen to melt the TiO_x surface layer of the Ti foam, and 3.0 ps with a time step of 1.0 fs was given to optimize the configuration. The structure obtained from previous AIMD simulation at 1500 K was used as the input structure for 298 K, and ran for 10 ps to fully reach its equilibrium configuration

using the gamma point of the Brillouin zone. The cutoff energy of the plane-wave basis was 400 eV, and the convergence accuracy of total energy for each step was less than 1×10^{-5} eV per atom in the AIMD simulation.

The relevant description was added as Supplementary Notes 4 on pages 48–49 in the revised supplementary information.

5. The author needs to check for other typo errors, such as in Supplementary Figure 17 line 127, where it should be "1M KOH" instead of "1M KCl".

Response: Thank you for your valuable comment. We carefully reviewed and revised the manuscript to correct the typo mentioned in Supplementary Figure 17. We also conducted a thorough check of the entire manuscript and supplementary materials to ensure there were no further typo errors.

Reviewer #2: In this manuscript, the authors address the critical issue of poor N₂ selectivity in the urea oxidation reaction (UOR) by constructing atomically isolated asymmetric Ni–O–Ti sites on commercial Ti foam. The asymmetric sites alter the urea adsorption configuration, thereby preventing C=N cleavage and significantly enhancing N₂ selectivity. The resulting anode with Ni–O–Ti sites demonstrates robust performance in actual urine denitrification and is utilized in a UOR-HER prototype for efficient solar-to-H₂ conversion. The specific comments are as follows:

1. From a theoretical perspective, why does the Ti foam containing symmetric Ti–O–Ti appear relatively inert for UOR?

Response: Thanks a lot for this constructive suggestion. During the revision, we performed DFT simulations to investigate the urea adsorption and the deprotonation step on symmetric Ti–O–Ti sites. Compared to urea adsorption via NH₂ groups on Ni–O–Ni sites (–0.46 eV, Supplementary Fig. 40a), urea prefers to the adsorption configuration on Ti–O–Ti sites via C=O group (–0.59 eV, Supplementary Fig. 40b), which is attributed to the high oxygenophilic affinity of Ti. However, the formation of *sp*³-hybridized intermediates on Ti–O–Ti, through bounding with the C moiety of the C=O group similar with the case of Ni–O–Ti, is very difficult due to the required energy input of 0.84 eV. Without the introduction of Ni single atoms to generate high valent Ni(III), the energy requirement for the initial dehydrogenation step of urea is extremely high, no matter urea adsorption at Ti–O–Ti sites via NH₂ groups (0.86 eV) or C=O groups (1.45 eV). Therefore, the symmetric Ti–O–Ti sites were inert for UOR.

During the revision, Supplementary Fig. 40 and its related discussion were added on page 41 in the revised supporting information.

Supplementary Figure 40. The Gibbs free energy change and optimized structures of urea adsorption and the first deprotonation step at the symmetric Ti–O–Ti sites of freestanding Ti foam via interactions with (a) the C=O group and (b) NH₂ groups.

2. The electrochemical surface area (ECSA) is crucial in the electrocatalytic process and should be measured, as it may influence the UOR activities of Ni–O–Ni and Ni–O–Ti sites.

Response: We thank the reviewer very much for this valuable comment. We estimated the electrochemical active surface area (ECSA) of asymmetric Ni–O–Ti and symmetric Ni–O–Ni sites by electrochemical double-layer capacitance (C_{dl}) measurements. The C_{dl} was analyzed from cyclic voltammetry measurements in a range of 1.00–1.10 V_{RHE} without Faraday current with a scan rate from 20 to 100 mV s⁻¹. By plotting the half of difference value in current density between anodic and cathodic sweeps ($\Delta j/2$) at a fixed potential of 1.05 V_{RHE} against the scan rate, the value of the fitting slope is C_{dl} . ECSA was calculated according to the equation:

$$ECSA = C_{dl}/C_s$$

C_s is the specific capacitance of sample or the capacitance of an atomically smooth planar surface of the material per unit area under identical electrolyte conditions. To estimate the surface

area, we used general specific capacitances of $C_s = 0.040 \text{ mF cm}^{-2}$ in 1.0 M KOH according to typical reported values (*Adv. Mater.* **2023**, 35, 2209338). As shown in Supplementary Fig. 11 a–c, the Ti foam with Ni–O–Ti sites had smaller C_{dl} value of 2.65 mF cm^{-2} than that of Ni–O–Ni sites (4.90 mF cm^{-2}). Therefore, the ECSA of Ni–O–Ti and Ni–O–Ni were 66.25 and 122.5 cm^2 , respectively. We further normalized UOR activity using ECSA, and found that asymmetric Ni–O–Ti sites had a much higher intrinsic UOR activity than symmetric Ni–O–Ni sites (Supplementary Fig. 11d).

During the revision, Supplementary Fig. 11 and its related discussion were added on page 12 in the revised supporting information.

Supplementary Figure 11. CV curves recorded in a range of 1.00–1.10 V_{RHE} at different scan rates of 20, 40, 60, 80, 100 mV s^{-1} for (a) asymmetric Ni–O–Ti and (b) symmetric Ni–O–Ni sites. (c) Half of current density differences ($\Delta j/2$) plotted against scan rates derived from CV curves, and (d) the

ECSA normalized LSV of UOR on asymmetric Ni–O–Ti and symmetric Ni–O–Ni sites.

3. The potential dissolution of Ni from the Ti foam into the aqueous solution during UOR, due to oxidation of Ni to a high valence state, should be investigated.

Response: We thank the reviewer very much for this valuable suggestion. During revision, we detected the amount of potential dissolved Ni in aqueous solution with Thermo Scientific iCAP™ Q ICP-MS during the durability test with the initial current density of 100 mA cm^{-2} in 1.0 M KOH containing 1.0 M urea (Supplementary Fig. 27). It was found that the Ni dissolution from asymmetric Ni–O–Ti and symmetric Ni–O–Ni was less than $1 \text{ } \mu\text{g cm}^{-2}$ after 24 hours of reaction, which was much lower than the theoretical Ni loading amount of Ni–O–Ti ($1200 \text{ } \mu\text{g cm}^{-2}$) and Ni–O–Ni ($4800 \text{ } \mu\text{g cm}^{-2}$) and ruled out the potential dissolution of Ni from the Ti foam.

During the revision, Supplementary Fig. 27 and its related discussion were added on page 28 in the revised supporting information.

Supplementary Figure 27. The amount of Ni dissolved from asymmetric Ni–O–Ti and symmetric Ni–O–Ni sites during 24 hours durability test with the initial current density of 100 mA cm^{-2} in 1.0 M

KOH containing 1.0 M urea.

4. The stability of the Ti foam with Ni single atoms after a long-term electrochemical reaction should also be examined.

Response: We thank the reviewer very much for this valuable suggestion. During the revision, we compared the high-resolution Ni 2p and Ti 2p XPS spectra of original and long-term used Ti foam loaded with Ni single atoms, and did not observe any obvious change of Ni 2p and Ti 2p XPS spectra after long-term electrochemical reaction (Supplementary Fig. 28). These results confirmed the excellent stability of Ni single atoms and Ti substrate during the reaction.

During the revision, Supplementary Fig. 28 and its related discussion were added on page 29 the revised supporting information.

Supplementary Figure 28. High-resolution (a) Ni 2p and (b) Ti 2p XPS spectrum of original and long-term used Ti foam with atomically isolated asymmetric Ni-O-Ti sites.

5. The use of different urea concentrations in linear sweep voltammetry (0.33 M) and stability tests (1.0 M) raises questions about the activity of the Ni-O-Ti sites under varying urea concentrations.

Response: We thank the reviewer for this valuable comment. During the revision, we measured the

UOR activities of Ni–O–Ti sites by LSV measurements in 1.0 M KOH containing different urea concentrations (Supplementary Fig. 12). The current density of UOR gradually increased with increasing urea concentration up to 0.33 M, and then kept unchanged at higher concentrations. Specifically, potentials of 1.34 and 1.33 V were respectively required to achieve a current density of 100 mA cm^{-2} at the urea concentrations of 0.05 and 0.10 M, and potential of 1.32 V was required at the urea concentrations of 0.33, 0.50, and 1.00 M.

During the revision, Supplementary Fig. 12 and its related discussion were added on page 13 in the revised supporting information.

Supplementary Figure 12. (a) The LSV curves of atomically isolated asymmetric Ni–O–Ti in 1.0 M KOH containing different urea concentrations. (b) The corresponding potentials of Ni–O–Ti for achieving a current density of 100 mA cm^{-2} at different urea concentrations.

6. The methods for detecting urea and total nitrogen concentrations should be specified in the Methods section of the main text.

Response: We thank the reviewer for this valuable reminder. Urea was quantified by high-performance liquid chromatography (HPLC, Thermo Scientific™ UltiMate™ 3000) method using a mobile phase

of acetonitrile–water (40:60, V/V) with a flow rate of 1.0 mL min⁻¹. The HPLC column was Agilent ZORBAX SB-C18 (4.6 × 150 mm). The detection wavelength was 195 nm. The injection volume was 10 μL.

The N-total concentration was determined by using an UV spectrophotometry according to the Chinese Standard HJ636–2012 (*Water quality–determination of total nitrogen-alkaline potassium persulfate digestion UV spectrophotometric method. Ministry of environmental protection of the People's Republic of China, China Environmental Science Press: Beijing, China, 2012.*). Briefly, the nitrogen of nitrogen compounds was converted into nitrate first by digestion using alkaline potassium persulfate for 30 min at 120 °C. The absorbance was measured at wavelengths of 220 nm and 275 nm after acidification with concentrated HCl. The corrected absorbance was equal to the absorbance at the wavelength of 220 nm minus twice the absorbance at 275 nm, which was proportional to the content of total nitrogen.

The above description was added in the Methods section on page 18–19 in the revised manuscript.

7. The authors should provide the k -space of the extended X-ray absorption fine structure (EXAFS) fitting results for both asymmetric Ni–O–Ti and symmetric Ni–O–Ni sites to show the reasonableness of the fitting results.

Response: Thanks a lot for your kind suggestion. During revision, we provided the raw data and fitted curves of EXAFS oscillations at k^3 -weighted Ni K-edge for asymmetric Ni–O–Ti and symmetric Ni–O–Ni sites (Supplementary Fig. 8). The fitting curve of k space was in good agreement with the original data, thus confirming the reasonableness of EXAFS fitting results.

During the revision, Supplementary Fig. 8 and its related discussion were added on page 9 in the revised supporting information.

Supplementary Figure 8. EXAFS oscillations at k^3 -weighted Ni K-edge and fitted curves of asymmetric Ni-O-Ti and symmetric Ni-O-Ni sites.

8. As shown in Fig. 5b, what is the calculation method for energy-saving efficiency in H₂ production?

Response: Thank you very much for this valuable comment. The energy-saving efficiency (η) in H₂ production was calculated according to the following equation:

$$\eta = (E_{\text{OER}} - E_{\text{UOR}})/E_{\text{OER}}$$

E_{OER} and E_{UOR} represent the potentials that reach a specific current density for OER and UOR, respectively. For a current density of 100 mA cm⁻² for Ni-O-Ti, the potentials required for OER and UOR were 1.61 and 1.33 V_{RHE}, respectively. Therefore, the energy-saving efficiency in H₂ production by Ni-O-Ti was 17% according to the above equation.

The relevant description was added as Supplementary Notes 2 on page 46 in the revised Supplementary information.

Reviewer #3: In this manuscript, Zhan et al. report an interesting single-atom electrochemical approach for urea pollutant control and resource utilization based on high selective urea oxidation reaction (UOR) coupled hydrogen evolution reaction (HER). They found that atomically isolated asymmetric Ni–O–Ti sites could prevent the generation of harmful byproducts (NCO^- or NO_x^-) and delivered an impressive N_2 selectivity of 99% during UOR with improved stability. The conceptualization and engineering of a prototype UOR-HER device, powered by commercial silicon photovoltaic cells, highlights an avenue for urea control and decentralized hydrogen production. Therefore, I recommend its publication in Nature Communications after some minor revisions. Specific comments are as follows.

1. How did the strong $\text{C=O} \rightarrow \text{Ti}$ interaction contribute to urea oxidation and N_2 evolution? Could urea be adsorbed on freestanding Ti foam?

Response: We thank the reviewer very much for these kind comments. The strong $\text{C=O} \rightarrow \text{Ti}$ interaction enables the O-terminal adsorption configuration of urea on Ni–O–Ti sites, which increases the electron density of the O atom and improves the resonance between the NH_2 and C=O group. The improved resonance reinforces the double bond character and bond strength of C=N bond, avoiding the urea cleavage into NCO^- and NH_x , and thus boosting the N_2 selectivity of urea oxidation.

The FTIR spectrum of urea on Ti foam was similar with the adsorption of urea on Ni–O–Ti sites (Supplementary Fig. 35), with a blueshift of $\nu_a(\text{C=N})$ frequency, suggesting that urea could be adsorbed on freestanding Ti foam.

During the revision, Supplementary Fig. 35 and its related discussion were added on page 36 in the revised supporting information.

Supplementary Figure 35. The FTIR spectra of urea adsorbed on freestanding Ti foam. The inset shows the possible adsorption configuration of urea on Ti–O–Ti sites. The inset shows the possible adsorption configuration of urea on Ti–O–Ti sites.

2. Please provide the electronic state of Ni single atoms after UOR to assess the stability of the electrode.

Response: Thank you for this valuable comments. During the revision, we measured the electronic state of Ni single atoms after UOR using high-resolution Ni 2p XPS, and did not observe any obvious change of Ni 2p XPS spectrum after the long-term reaction. This result confirmed the excellent stability of Ni single atoms during UOR (Supplementary Fig. 28).

During the revision, Supplementary Fig. 28 and its related discussion were added on page 29 in the revised supporting information.

Supplementary Figure 28. High-resolution Ni 2p XPS spectrum of original and long-term used Ti foam with atomically isolated asymmetric Ni–O–Ti sites.

3. Does the preparation temperature influence the UOR performance of Ni–O–Ti sites?

Response: We thank the reviewer very much for this constructive comment. During the revision, we adjusted the preparation temperatures to 200 °C, 300 °C, 400 °C, and 500 °C to construct the asymmetric Ni–O–Ti sites without changing other conditions, and measured their UOR performances by LSV curves in 1.0 M KOH + 0.33 M urea (Supplementary Fig. 10). The activity of Ni–O–Ti sites increased when increasing the preparation temperature from 200 °C to 300 °C, and then slightly decreased with the temperature increasing from 300 °C to 500 °C. Specifically, the UOR potential at a current density of 10 mA cm^{-2} was $1.32 \text{ V}_{\text{RHE}}$ at the preparation temperature of 200 °C, and decreased to $1.30 \text{ V}_{\text{RHE}}$ when the temperature increased to 300 °C, 400 °C, and 500 °C. For the current density up to 100 mA cm^{-2} , the UOR potentials for the electrodes prepared at temperature of 200 °C, 300 °C, 400 °C, and 500 °C were 1.38, 1.32, 1.33, and $1.33 \text{ V}_{\text{RHE}}$, respectively.

During the revision, Supplementary Fig. 10 and its related discussion were added on page 11 in the revised supporting information.

Supplementary Figure 10. The LSV curves of Ti foam with atomically isolated asymmetric Ni–O–Ti sites under different preparation temperature of 200 °C, 300 °C, 400 °C, and 500 °C in 1.0 M KOH + 0.33 M urea. (b) The corresponding potentials of Ni–O–Ti sites with different preparation temperature at current density of 10 mA cm⁻² and 100 mA cm⁻², respectively.

4. The specific contribution of Ni–O–Ti shells in FT-EXAFS R-space should be determined to support Ni–O–Ti coordination more clearly.

Response: Thanks a lot for this valuable comment. During the revision, we analyzed the FT-EXAFS spectrum of Ni–O–Ti sites by using two backscattering paths of Ni–O and Ni–O–Ti (Supplementary Fig. 7). Curves from top to bottom were the Ni–O–Ti and Ni–O two-body backscattering signals χ_2 included in the fit and the total signal (red line) superimposed on the experimental signal (black dots). The measured and calculated spectra agreed well. The best-fitting analyses revealed that the main peak was originated from Ni–O first shell coordination and the minor peak was well interpreted as the Ni–O–Ti contribution.

During the revision, Supplementary Fig. 7 and its related discussion were added on page 8 in

the revised supporting information.

Supplementary Figure 7. The specific contribution of Ni–O–Ti and Ni–O shells coordination in FT-EXAFS for Ni–O–Ti sites. The inset displayed the structure of a Ni–O–Ti moiety derived from the EXAFS result, where the blue, red, and grey spheres represented Ni, O and Ti, respectively.

5. The charge transfer resistance of Ti foams with Ni–O–Ti and Ni–O–Ni sites may affect their UOR activity, which should be measured using electrochemical impedance spectroscopy.

Response: Thanks a lot for the insightful comment. During the revision, the electrochemical impedance spectroscopy (EIS) measurements were performed at a frequency range from 100 kHz to 0.1 Hz. The EIS plots were fitted using simplified Randles circuit model, in which the R_{ct} represented the charge transfer resistance. The experimental results revealed that R_{ct} of Ni–O–Ti (73 Ω) was much smaller than that of Ni–O–Ni (138 Ω), indicating a favorable charge transfer kinetics of Ni–O–Ti.

During the revision, Supplementary Fig. 13 and its related discussion were added on page 14 in the revised supporting information.

Supplementary Figure 13. The EIS spectra of symmetric Ni–O–Ni and asymmetric Ni–O–Ti sites at 1.30 V_{RHE} . Inset shows the equivalent electric circuit. The EIS plots were fitted using simplified Randles circuit model for obtaining the series resistance (R_s), capacitance of the double layer (C_d) and charge-transfer resistance (R_{ct}).

6. Were Nafion membranes applied to treat actual urine in the UOR-HER device? The utilization of Nafion membranes may increase the system resistance.

Response: Thanks a lot for these insightful comments. We used Nafion 117 membrane to treat actual urine in the UOR-HER device. Although the Nafion membrane increased the resistance and reduced the current density of the system (Supplementary Fig. 43a), it separated the generated H_2 and avoided its oxidation at the anode, thus improving the efficiency of hydrogen production (Supplementary Fig. 43b).

During the revision, Supplementary Fig. 43 and its related discussion were added on page 44 in the revised supporting information.

Supplementary Figure 43. (a) The UOR polarization curves with and without Nafion membrane using asymmetric Ni–O–Ti sites as the anode and Pt foil as the cathode, and (b) corresponding hydrogen production rate.

7. Some very recent literatures about UOR should be cited, such as *Adv. Funct. Mater.*, 2023, 2303986; *J. Am. Chem. Soc.*, 2022, 144, 1174–1186; *Chem Catalysis* 2024, 4, 100840.

Response: Thank you for this valuable suggestion. Related references have been cited in the revised manuscript as follows.

Ref. 7. *Adv. Funct. Mater.* **33**, 2303986 (2023);

Ref. 8. *Chem Catal.* **4**, 100840 (2024);

Ref. 41. *J. Am. Chem. Soc.* **144**, 1174–1186 (2022);

Ref. 50. *Sci. Bull.* **67**, 1763–1775 (2022).

REVIEWERS' COMMENTS

Reviewer #1 (Remarks to the Author):

The revised version has fully addressed the suggestions of the reviewers. It is currently acceptable.

Reviewer #2 (Remarks to the Author):

It is can be accepted now.

Reviewer #3 (Remarks to the Author):

All my questions and concerns have been well addressed. It can be accepted as is.